# When Code Authors Are Agents: A Large-Scale Study of Human–Agent Collaboration in Pull Requests

## Abstract

Autonomous coding agents are increasingly participating in collaborative software development by generating repository-level pull requests (PRs) that must be reviewed and integrated by human teams. While prior work has examined the technical characteristics of agent-generated patches, less is known about how autonomous authorship reshapes human collaboration dynamics in real-world workflows. In this paper, we investigate large-scale human–agent collaboration by comparing 40,214 pull requests across 2,807 GitHub repositories, including 33,596 agent-authored PRs from five autonomous coding agents and 6,618 human-authored PRs. We examine differences across three dimensions: integration outcomes, structural characteristics, and collaboration signals.

Our findings reveal a socio-technical trade-off. Agent-authored PRs are integrated significantly faster, yet exhibit lower merge rates overall. Task type moderates this effect: agents outperform humans in documentation tasks but underperform in behavior-changing contributions. Beyond outcomes, collaboration patterns differ systematically. Agent-authored PRs attract proportionally more bot-generated comments and elicit more analytic, less socially oriented review communication. In contrast, human-authored PRs receive more elaborative and socially engaged feedback.

Incorporating psycholinguistic features into predictive models significantly improves merge outcome prediction, demonstrating that communication style carries explanatory power beyond structural code characteristics. These results suggest that autonomous agents do not merely introduce technical artifacts into repositories, but actively reshape review interaction patterns and evaluative behavior. The impact of coding agents is therefore fundamentally socio-technical, highlighting the importance of studying AI systems within authentic human–AI collaborative environments.

## CCS Concepts

• **Human-centered computing** → *HCI theory, concepts and models*; • **Computing methodologies** → **Intelligent agents**; • **Software and its engineering** → **Open source model**.

## Keywords

AI coding agents, Human–AI interaction, Code review dynamics, Collaboration signals

**ACM Reference Format:**
Anonymous Author(s). 2018. When Code Authors Are Agents: A Large-Scale Study of Human–Agent Collaboration in Pull Requests. In *Proceedings of Make sure to enter the correct conference title from your rights confirmation email (Conference acronym 'XX).* ACM, New York, NY, USA, 8 pages. https://doi.org/XXXXXXX.XXXXXXX

## 1 Introduction

The landscape of software development is evolving with the increasing integration of AI-powered tools. We are witnessing a shift from interactive coding assistants to more autonomous AI agents [1, 5]. Coding assistants operate within a developer's workflow, offering suggestions that can be accepted, revised, or ignored in real time, preserving clear human authorship and decision control. In contrast, autonomous agents such as OpenAI Codex [23] and Anthropic's Claude Code Agent [2] act at the repository level, generating complete pull requests and participating directly in collaborative review processes. This shift represents not merely a technical evolution, but a reconfiguration of agency within socio-technical workflows.

As AI systems move from suggestion-level support to repository-level authorship, they become active contributors rather than background assistants. Their outputs are no longer fragments embedded within human activity, but cohesive change proposals that must be interpreted, evaluated, negotiated, and either accepted or rejected by human teams. This transition raises fundamentally interactional questions. How do developers interpret AI-authored contributions compared to those of human peers? Does perceived authorship shape scrutiny, explanation, and trust? Broadly, how does the presence of autonomous contributors subtly reshape collaborative norms?

In contemporary software development, pull requests (PRs) constitute the primary arena in which such questions become observable. A PR is not merely a code submission. It is a structured interaction space where technical artifacts and social evaluation intersect. Contributors propose changes. Reviewers interpret them. Discussion unfolds. Integration decisions are made. When autonomous agents operate at the repository level, their participation materializes precisely through PR. Therefore, pull requests provide a natural empirical lens for studying human–AI interaction in large-scale collaborative development.

Prior work shows that PR rejection is often driven not only by code defects but by coordination challenges, scope misalignment, and integration constraints [8, 21]. Research on agent-generated patches suggests that structural organization and change distribution, rather than raw size alone, distinguish autonomous contributions [22]. At the same time, studies on trust in code review indicate that perceptions of authorship can subtly shape scrutiny and evaluative behavior, even when objective quality is comparable [34]. Together, these findings suggest that repository-level autonomy may reshape not only what changes are introduced, but how those changes are interpreted, discussed, and integrated. Yet, large-scale

empirical evidence comparing human- and agent-authored pull requests across integration outcomes, structural characteristics, and review interaction patterns remains limited.

In this paper, we investigate these human–AI interaction dynamics at scale by comparing agent-authored and human-authored pull requests across development outcomes, structural characteristics, and collaboration signals. We conduct a large-scale empirical study of 40,214 pull requests across 2,807 GitHub repositories, including 33,596 agent-authored PRs from five autonomous coding agents and 6,618 human-authored PRs.

Our findings reveal a consistent trade-off. Agent-authored pull requests are integrated significantly faster, yet exhibit lower merge rates overall. Task type moderates this effect, with agents outperforming humans in documentation tasks but underperforming in behavior-changing contributions. Beyond outcomes, structural signals and review language also differ systematically. Agent-authored pull requests exhibit higher complexity and churn despite comparable breadth, attract more bot-generated comments, and elicit more analytic, less socially oriented review communication.

This paper makes three contributions. First, we provide the first large-scale empirical comparison of human and autonomous authorship in open-source pull requests. Second, we demonstrate that repository-level autonomy reshapes not only integration outcomes but also how contributions are structurally interpreted and communicatively processed. Third, we show that linguistic interaction patterns offer additional explanatory power for merge decisions, highlighting that the impact of coding agents is fundamentally socio-technical rather than purely technical.

## 2 Design

This study investigates how agentic code authors differ from human developers across three complementary dimensions of software development: (i) outcome-level performance, (ii) patch-level code quality, and (iii) collaboration and communication signals. These dimensions jointly frame our following research questions and analytical approach.

**RQ1.** How do development outcomes differ between agent-authored and human-authored pull requests?

**RQ2.** How do patch characteristics differ between agent-authored and human-authored pull requests?

**RQ3.** How do collaboration and communication patterns differ in agent–human versus human–human code review interactions, and how do these patterns relate to merge outcomes?

### 2.1 Data Mining

This paper uses the AIDEV-pop dataset, a subset of the AIDEV dataset [17], which is a large-scale dataset capturing the activities of autonomous coding agents on GitHub repositories when generating pull requests. The dataset provides metadata on pull requests (PRs), including authorship, review timelines, code changes, and integration outcomes. From AIDEV-pop, we retained only repositories with more than 100 stars. To enable robust comparisons with human-authored PRs, we also constructed a Human-PR dataset sampled from the same repositories as the agentic PRs, restricting to repositories with more than 500 stars. This final subset contains 33,596 PRs authored by five autonomous coding agents (OpenAI

Codex, Devin, GitHub Copilot, Cursor, and Claude Code) across 2,807 repositories, and 6,618 human-authored PRs, resulting in 40,214 unique PRs in total.

Following the workflow illustrated in Figure 1, we augmented, merged, and preprocessed the data prior to analysis. All analyses were conducted at the PR level (N = 40,052) for RQ1 and RQ3, and at the patch level for RQ2 (N = 999,579), excluding PRs without comments for the psycholinguistic analyses.

### 2.2 Measures

We operationalize our research questions using measures that capture integration outcomes, structural properties of pull requests, and collaboration dynamics. These dimensions allow us to examine both the technical and interactional aspects of agent and human contributions.

#### 2.2.1 RQ1: Outcome-Level Performance.
At the outcome level, we focus on whether agent-authored pull requests (PRs) are accepted and how quickly they are integrated. We characterize development outcomes using two measures: (i) *merge rate*, defined as the proportion of merged PRs among all submitted PRs, and (ii) *time-to-merge*, measured as the elapsed time between PR creation and merge, computed for merged PRs only.

These metrics capture both the likelihood of acceptance and the efficiency of review and integration, which are central indicators of practical usefulness in collaborative software development.

#### 2.2.2 RQ2: PR characteristics.
To assess patch quality, we analyze multiple dimensions of change content and structure. These metrics capture not only how much code is modified, but also the functional intent and structural properties of the proposed change, which prior work has shown to influence review effort, integration likelihood, and coordination complexity [8, 21]. Specifically, we consider:

- **Task type**, capturing the functional intent of a PR.
- **Breadth of change**, measured by the number of files changed and the number of hunks, reflecting the dispersion of modifications across the codebase.
- **Size of change**, measured using code churn (additions + deletions) and diff size, where diff size reflects the number of logical changes rather than raw line counts.
- **Structural complexity**, measured via cyclomatic complexity, computed using Lizard [15, 35], a multi-language static analysis tool suitable for heterogeneous PR patches.
- **Discussion volume**, captured by the number of comments, commits, and reviews associated with each PR, serving as a proxy for review effort and coordination intensity.

The AIDEV dataset contains 12 fine-grained task labels (`feat` (feature addition), `fix` (bug fix), `docs` (documentation), `build`, `ci` (continuous integration), `refactor`, `test`, `perf` (performance improvements), `style`, `chore` (routine maintenance), `revert`, and `other`). The revert and other types were removed as they were significantly lower across PRs compared to other task types. To enable meaningful comparisons, we grouped these into four higher-level task categories based on functional similarity:

- **Feature & Behavior-changing**: feat, fix, perf
- **Documentation & Style**: docs, style

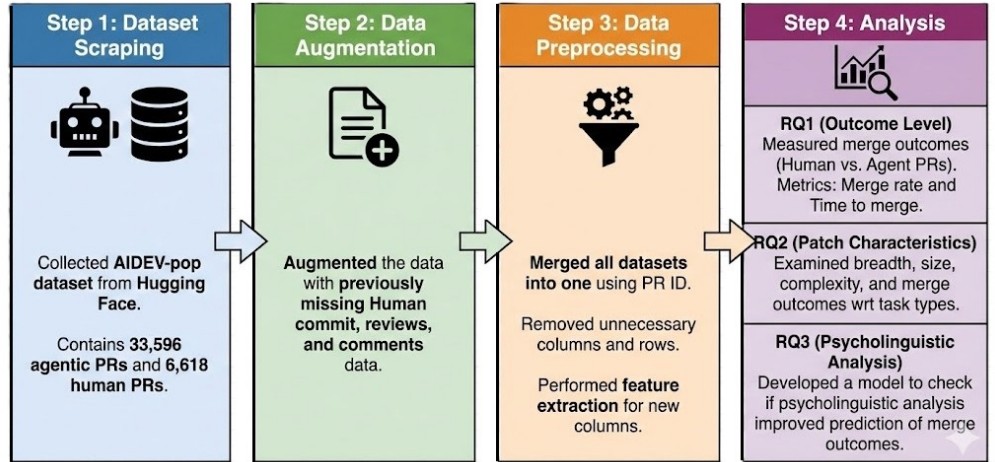

**Figure 1: Workflow of our large-scale human–AI pull request study, detailing data acquisition, enrichment with collaboration signals, preprocessing and feature engineering, and multi-level analysis of integration outcomes and interaction patterns.**

- **Infrastructure & Automation**: build, ci, chore
- **Internal Code Quality**: refactor, test

*2.2.3 RQ3: Collaboration and Communication Signals.* Beyond structural code characteristics, successful software development is fundamentally shaped by human collaboration. To capture these interactional dynamics, we analyze review and discussion comments associated with pull requests. Specifically, we examine: (i) the distribution of human versus bot commenters, and (ii) the psycholinguistic properties of review language.

We employ Linguistic Inquiry and Word Count (LIWC)-based [31] psycholinguistic analysis to quantify dimensions such as tone, analytic thinking, social orientation, clout, and emotional expression in reviewer comments. This approach is well established in human factors and software engineering research and allows us to examine how reviewers adapt their communication style when engaging with agent-authored versus human-authored contributions.

The extracted features are categorized into Style Indicators and Psychological Processes (Table 1).

## 3 Data Analysis and Results

In this section, we report empirical results for each research question. We present descriptive statistics at the pull request level and evaluate differences between agent-authored and human-authored PRs using Chi-square tests, MannWhitney U, correlation analyses, and logistic regression models. Effect sizes are reported using odds ratios, regression coefficients, and correlation coefficients. For predictive models, we assess improvements in model fit using likelihood ratio tests and information criteria (AIC/BIC).

### 3.1 RQ1: Outcome-Level Performance

We examine outcome-level performance differences between human and agent authors, focusing on merge rate and time to merge as illustrated in Table 2.

Human-authored PRs show a significantly higher merge rate than agent-authored PRs (77.2% vs. 71.5%; $\chi^2$, $p < 0.001$, $V = 0.05$).

**Table 1: Linguistic and psychological features derived from LIWC to analyze stylistic and interactional properties of pull request review comments.**

| Dimension | Description |
|---|---|
| **Style Indicators (High-Level Summary)** | |
| Tone | Overall emotional valence of the message. |
| Analytic | Degree of logical or analytical reasoning expressed. |
| Clout | Confidence or dominance conveyed in the text. |
| Authentic | Degree of personal or self-revealing language. |
| Word Count | Total number of words in the message. |
| **Psychological Processes** | |
| polite | Expressions of politeness, courtesy, or respect. |
| insight | Language indicating understanding or realization. |
| cause | References to causal reasoning or explanations. |
| emo_anger | Language expressing anger or hostility. |
| emo_anx | Language expressing anxiety or worry. |
| Social | References to social interaction or relationships. |

In contrast, a Mann-Whitney U test indicated that agent-authored PRs are merged substantially faster once submitted, with a mean time-to-merge of 19.43 hours compared to 63.89 hours for human-authored PRs. This difference in time-to-merge is also statistically significant ($p < 0.001$, $r = -0.55$).

Overall, the results highlight a trade-off between acceptance likelihood and integration speed across author types. Human-authored contributions are more likely to be accepted, whereas agent-authored contributions move through review and integration more quickly when they are merged.

### 3.2 RQ2: PR characteristics

To better understand whether merge outcomes vary across types of work, we examine merge rates by task category and author type. This analysis allows us to assess whether the relative performance

**Table 2: Comparison of merge outcomes and review latency for human- and agent-authored pull requests. Human-authored PRs exhibit a higher merge rate, while agent-authored PRs show substantially shorter average time-to-merge (hours).**

| Author | Total PRs | Merged PRs | % Merged | Time-to-Merge(hr) |
|---|---|---|---|---|
| Human | 6503 | 5019 | 77.2 | 63.89 |
| Agent | 33549 | 23999 | 71.5 | 19.43 |

**Table 3: Task-level comparison of merge outcomes for agent- and human-authored pull requests, highlighting variation in integration likelihood across change types. Agents outperform humans in documentation tasks, while humans achieve higher merge rates in behavior-changing contributions.**

| Task Category | Author | Total | Merged | Merge Rate |
|---|---|---|---|---|
| Docs & Style | Agent | 4,075 | 3,413 | 83.8 |
| | Human | 578 | 450 | 77.9 |
| Feature & Behavior | Agent | 22,896 | 15,725 | 68.7 |
| | Human | 3,812 | 2,952 | 77.4 |
| Infrastructure | Agent | 1,934 | 1,410 | 72.9 |
| | Human | 1,561 | 1,202 | 77.0 |
| Internal Quality | Agent | 4,644 | 3,451 | 74.3 |
| | Human | 552 | 415 | 75.2 |

of agents and humans depends on the nature of the task, rather than treating all pull requests (PRs) as homogeneous. Table 3 reports merge rates by task category and author type.

*3.2.1 Task Categories and Merge Outcomes.* Merge rates vary substantially across task categories. Agent-authored PRs achieve their highest merge rate in Docs and Style (83.8%), exceeding human-authored PRs in this category (77.9%). In contrast, humans outperform agents in more code-centric categories, including Feature and Behavior (77.4% vs. 68.7%), Infrastructure (77.0% vs. 72.9%), and Internal Quality (75.2% vs. 74.3%). The gap is particularly pronounced for Feature and Behavior tasks, which represent the largest share of PRs. A logistic regression model predicting merge likelihood reveals a significant interaction between author type and task category ($\chi^2$, $p < 0.001$), indicating that the effect of author type depends on the nature of the task. Specifically, humans are significantly more likely to have PRs merged in Feature and Behavior ($\beta = 0.90$, $p < 0.001$), Infrastructure ($\beta = 0.55$, $p < 0.001$), and Internal Quality ($\beta = 0.48$, $p < 0.001$). In contrast, agents retain an advantage in documentation-related tasks ($\beta_{\text{human}} = -0.35$, $p < 0.001$). Figure 2 visualizes this interaction between author type and task category, in terms of merge outcomes.

*3.2.2 Breadth of Change.* Regarding the breadth of changes, human-authored PRs modify a larger portion of the codebase. Humans change more files (median 4 vs. 3) and modify more hunks per PR (median 9 vs. 7), with both differences statistically significant (Mann-Whitney U, $p < 0.001$).

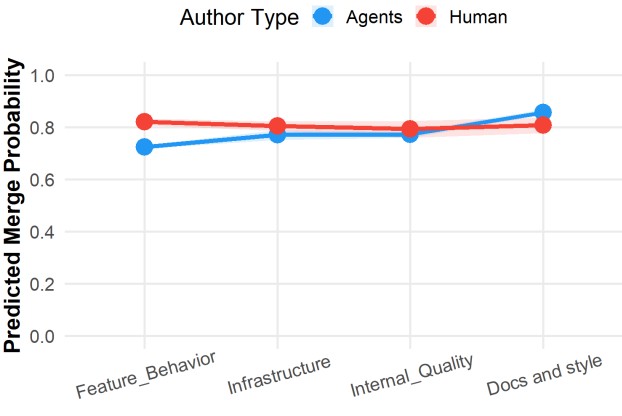

**Figure 2: Model-predicted merge probabilities by task category, showing agents outperforming in documentation tasks and underperforming in behavior-changing tasks.**

*3.2.3 Size of Change.* We find no statistically significant differences in the overall size of changes between agent- and human-authored PRs. Specifically, neither diff size (Mann–Whitney U, $p = 0.21$) nor code churn (Mann–Whitney U, $p = 0.22$) differs significantly across the two groups, suggesting comparable volumes of code modification.

Together, these results suggest that while agents introduce changes of similar overall size, they tend to produce more structurally complex modifications concentrated within fewer files, whereas humans distribute changes more broadly across the codebase.

*3.2.4 Structural Complexity.* For structural complexity, agent-authored PRs exhibit significantly higher average cyclomatic complexity ($\mu = 1.85$) than human-authored PRs ($\mu = 1.33$; Welch's $t$-test, $p < 0.001$, $d = 0.27$), indicating more complex control-flow structures within individual patches.

For human-authored PRs, cyclomatic complexity is weakly but statistically significantly negatively correlated with merge success ($r = -0.06$, $p < 0.001$) and positively correlated with time-to-merge ($\rho = 0.24$, $p < 0.001$). Although the effect sizes are small, this pattern suggests that more complex human-authored patches require longer review cycles and are marginally less likely to be accepted.

In contrast, agent-authored PRs exhibit a weak positive association between complexity and merge success ($r = 0.05$, $p < 0.001$), along with a negligible negative correlation with time-to-merge. This indicates that, for agents, higher structural complexity does not impede integration and may even coincide with slightly higher acceptance rates. Importantly, all observed effect sizes are small ($|r| < 0.12$), indicating that cyclomatic complexity alone explains only a limited portion of the variance in merge outcomes. Complexity appears to play a secondary role relative to other factors in determining review efficiency and acceptance.

*3.2.5 Discussion Volume.* With a logistic regression model predicting merge likelihood, result shows that higher discussion volume

**Table 4: Comparison of LIWC Metrics in Reviewer Comments for Agent- and Human-Authored PRs**

| Metric | Agent | Human | Δ% | *p*-value | *d* |
|---|---|---|---|---|---|
| **Style Indicators** | | | | | |
| Tone | 61.06 | 65.62 | −7.0% | < 0.001 | −0.14 |
| Analytic | 56.79 | 54.19 | +4.8% | < 0.001 | 0.09 |
| Clout | 39.28 | 41.85 | −6.1% | < 0.001 | −0.08 |
| Authentic | 45.68 | 44.86 | +1.8% | 0.37 | 0.03 |
| Word Count | 49.19 | 63.59 | −22.6% | < 0.001 | −0.09 |
| **Psychological Processes** | | | | | |
| polite | 1.96 | 2.13 | −7.9% | < 0.001 | −0.02 |
| insight | 2.68 | 2.85 | −5.8% | < 0.001 | −0.03 |
| cause | 2.49 | 2.84 | −12.6% | < 0.001 | −0.09 |
| emo_anger | 0.02 | 0.02 | +8.3% | 0.02 | 0.01 |
| emo_anx | 0.01 | 0.02 | −35.8% | < 0.001 | −0.03 |
| Social | 8.73 | 9.62 | −9.2% | < 0.001 | −0.06 |

**Table 5: Distribution of bot and human commenters across agent- and human-authored pull requests, indicating greater bot engagement in agent-authored pull requests compared to human-authored ones.**

| Author Type | Bot (%) | Human (%) |
|---|---|---|
| Agent | 61.46 | 38.54 |
| Human | 51.42 | 48.58 |

(number of comments) is negatively associated with merge likelihood ($\beta = -0.14, p < 0.001$), whereas the number of reviews shows a positive association. This pattern suggests that prolonged discussion may reflect contentious or problematic PRs, while active reviewer engagement increases the likelihood of acceptance.

## 3.3 RQ3: Collaboration and Communication Signals in PRs

To examine differences in collaboration dynamics, we analyze both structural interaction patterns and the linguistic characteristics of reviewer comments.

*3.3.1 Commenter Type Distribution.* A chi-square test reveals a strong association between PR author type and commenter type ($\chi^2(1) = 1652.16, p < 0.001$), indicating that the distribution of comment sources varies systematically across author groups. Agent-authored PRs are significantly more likely to receive bot-generated comments, with an odds ratio of 1.51. This suggests that automated contributions attract proportionally more automated review activity, reflecting a partially automated interaction loop around agent-generated patches.

*3.3.2 Psycholinguistic Analysis.* Table 4 summarizes LIWC-based psycholinguistic differences in reviewer comments using the Mann-Whitney U test.

Reviewer comments on agent-authored PRs exhibit lower emotional tone and higher analytic style, indicating more formal and task-focused communication.

In contrast, comments on human-authored PRs contain higher word counts, greater social orientation, and higher clout, suggesting more conversational and elaborative interactions. Significant differences in the *Cause* metric further indicate that reviewers are more likely to provide explanatory reasoning when responding to human authors, whereas feedback directed at agents appears comparatively more concise and directive. Notably, no significant differences emerge for anger, suggesting the absence of overtly negative or hostile language toward agent-authored contributions.

To assess whether communication style contributes to merge outcomes, we incorporate psycholinguistic variables (Tone and Analytic) into merge prediction models. Compared to a base model excluding LIWC features, the extended model demonstrates improved fit, reflected in lower AIC and BIC values and a statistically significant likelihood ratio test ($\chi^2(2) = 49.28, p < 0.001$).

These results indicate that reviewer communication style provides incremental explanatory power beyond author type, task category, and code characteristics. Linguistic signals are therefore not merely descriptive of interaction tone but are meaningfully associated with integration outcomes.

## 4 Discussion

Our findings highlight meaningful socio-technical differences between agent-authored and human-authored contributions in open-source development. We synthesize the implications of each research question below.

**RQ1. Author Type and Development Outcomes.** Our results reveal a clear trade-off between acceptance likelihood and integration speed. Human-authored PRs exhibit higher merge rates, whereas agent-authored PRs are integrated significantly faster once merged. This pattern suggests that autonomous agents may operate efficiently within review pipelines but face a modest acceptance disadvantage relative to human contributors. The faster time-to-merge for agents may reflect narrower discussion cycles, automation-compatible changes, or reduced negotiation dynamics. Conversely, higher acceptance rates for human-authored PRs may indicate stronger alignment with repository norms or greater trust in human contributors. Together, these findings suggest that repository-level autonomy alters not only productivity dynamics but also evaluative thresholds in collaborative development.

Prior work reports that agentic PRs generally involve smaller, more localized changes than human PRs [22], which may contribute to faster integration. They also shows that rejection of AI-generated PRs is often driven by coordination and scope misalignment with project goals rather than code defects [8, 21]. However, variation across tools indicates that agent behavior is not uniform, highlighting the importance of considering differences between agents.

**RQ2. Author Type and PR Characteristics.** Differences between agent and human contributions extend beyond outcomes to the structural properties of changes. While size of change (diff size and churn) did not differ significantly, agent-authored PRs exhibited higher cyclomatic complexity and slightly more concentrated structural footprints. These findings indicate that agent-generated

patches may involve more complex control-flow changes within fewer files or regions, even when comparable in size. Importantly, task category moderates merge likelihood: agents outperform humans in documentation-related tasks, whereas humans retain advantages in feature and behavior-changing contributions. This interaction suggests that repository-level autonomy is better suited to certain functional domains, particularly those involving localized or stylistic updates, while more behaviorally impactful changes may still benefit from human authorship.

Ogenrwot *et al.* [22] show that agentic PRs differ across tools and are distinguished less by size than by how changes are structured and distributed, consistent with our finding that structural signals better differentiate author types than raw volume.

**RQ3. Collaboration and Communication Signals.** Autonomous authorship also shapes interaction patterns. Agent-authored PRs are significantly more likely to receive bot-generated comments, indicating a partially automated review loop surrounding agent contributions. Psycholinguistic analysis further reveals systematic differences in communication style: reviewer comments on agent-authored PRs exhibit higher analytic tone and lower social orientation, whereas comments on human-authored PRs are longer, more socially oriented, and more explanatory. Notably, no differences were observed in anger, and a negligible difference was found in anxiety, suggesting that agent contributions do not provoke overt hostility. Moreover, incorporating linguistic features into predictive models improves merge outcome prediction, indicating that communication style carries explanatory value beyond structural code characteristics.

As noted by Ogenrwot *et al.,* [22] the structure of agentic contributions may shape reviewer responses: tightly scoped edits tend to elicit directive feedback, while broader changes invite more explanatory discussion. Thus, communication patterns reflect not only authorship but also how changes are organized. In addition, Yabesi *et al.,* [34] shows that trust cues can influence developers' cognitive processing without necessarily changing their stated decisions. This suggests that differences in review dynamics and merge outcomes may partly reflect implicit trust calibration toward agent authorship, shaping levels of scrutiny and communication style even when overt acceptance decisions remain stable.

## 4.1 Implications for Research

This study contributes to emerging work on human–AI collaboration by demonstrating that autonomous coding agents participate in socio-technical systems in ways that measurably shape outcomes, structural signals, and communication dynamics. Unlike prior benchmark-based evaluations, our findings underscore the importance of studying AI systems within real-world collaborative contexts. The observed interaction between task type and author identity suggests that future research should move beyond aggregate performance metrics and examine domain-specific suitability of autonomous tools. Additionally, the predictive contribution of psycholinguistic features highlights the value of integrating human-centered language analysis into large-scale mining studies.

Future research should examine how these dynamics evolve over time as developers gain familiarity with agentic systems. Longitudinal analyses may reveal shifts in trust calibration, review behavior,

and structural acceptance norms. Furthermore, investigating hybrid workflows—where humans iteratively refine agent-generated patches—may provide deeper insight into effective human–AI co-development strategies.

## 4.2 Implications for Practice

For practitioners, our findings suggest that autonomous coding agents may be particularly effective for documentation and stylistic updates, where acceptance rates are competitive or superior. However, more behaviorally impactful changes may require additional human oversight or iterative refinement to achieve comparable acceptance levels. The faster integration time of agent-authored PRs indicates potential efficiency gains, especially in repositories with automated validation pipelines.

Differences in review communication patterns also suggest that agent contributions may benefit from enhanced explainability or contextual annotations to support more socially engaged review discussions. Designing AI tools that better surface rationale, intent, or traceability could reduce interpretive friction and improve integration outcomes.

As repository-level autonomy becomes more prevalent, understanding how structural, behavioral, and communicative signals interact will be critical for designing collaborative environments that support effective human–AI co-development.

## 5 Related Work

Our related work section covers three main areas. agentic development in software engineering, human–agent collaboration, and psycholinguistic analysis in software engineering, which together frame the technical and socio-cognitive foundations of this study.

## 5.1 Agentic Development

As AI-powered tools become increasingly integrated into software development workflows, developers are beginning to rely on more autonomous AI agents capable of performing end-to-end development tasks [13, 18]. This shift raises fundamental questions about the development styles of these agents, particularly whether and how their behavior converges toward human-like software development practices.

Prior research has examined agent-based approaches for specific development tasks, such as code generation [37, 40], code review [30], refactoring [11, 24], build maintenance [7], and automated program repair [39]. Recent studies explore agents capable of executing multi-step development processes and submitting pull requests (PRs) to open-source repositories [10, 26, 36], positioning agents as first-class contributors rather than passive assistants.

Focusing specifically on agent-authored pull requests, Watanabe *et al.* [33] and Pham and Ghaleb [26] report that such PRs are more frequently accepted for documentation-related tasks, while human-authored PRs remain more likely to be merged overall. Pham and Ghaleb [26] further observe similarities in the volume of code produced by human- and agent-authored PRs, indicating that differences in outcomes may stem less from patch size than from other structural or contextual factors. Milanese *et al.* [19] find that agent-authored PRs and human-authored PRs are similarly likely to include test-related changes. However, when tests are present,

human-authored PRs tend to modify a larger proportion of test files and lines of code.

More broadly, software engineering research on patch quality has examined characteristics such as patch size, cyclomatic complexity [12, 16], churn [4], and the number of modified files [12]. These metrics provide a foundation for assessing whether agent-authored changes differ substantively from human-authored ones.

Building on this literature, our work systematically compares agent- and human-authored pull requests along outcome-level performance and structural characteristics. Specifically, we quantify trade-offs between merge likelihood and integration speed and examine how task type, change breadth, and code complexity differentiate agent and human contributions. This provides large-scale empirical evidence on how agentic development differs from human development in contemporary open-source workflows.

## 5.2 Human-Agent Collaboration

As AI agents transition from assistive tools to autonomous contributors, software development increasingly involves collaboration between human developers and these agents. Prior work on Human-AI collaboration in software engineering emphasizes challenges related to trust, coordination, accountability, and role clarity when non-human actors participate in collaborative workflows [3, 14, 34]. These challenges are further amplified in open-source settings, where the integration of AI-assisted contributions reshapes how trust is formed and evaluated among collaborators [28].

Recent studies suggest that developers calibrate their interaction strategies based on the perceived capabilities and reliability of AI agents, often applying different levels of scrutiny or engagement compared to human contributors [25, 32]. As a result, agent authorship can reshape not only development outcomes but also collaboration dynamics, influencing how contributions are reviewed, discussed, and integrated. Our work builds on this perspective by empirically examining how agent participation affects both technical and social dimensions of collaborative development in practice.

## 5.3 Psycholinguistic Analysis

Psycholinguistic analysis has been widely used in software engineering research to systematically study developer communication patterns, collaboration quality, and socio-emotional dynamics in artifacts such as issue discussions, code reviews, and pull request comments [6, 27]. Linguistic features derived from these interactions have been shown to reliably reflect cognitive load, domain expertise, coordination effort, and trust-related cues in collaborative development settings [9, 20].

More recently, researchers have applied psycholinguistic methods to understand how developer behaviour and communication patterns shift in the presence of AI-generated artifacts [29, 38]. These studies suggest that linguistic signals can reveal implicit differences in how developers perceive and engage with AI-authored contributions, even when explicit decisions such as acceptance or rejection remain unchanged. In this work, we leverage psycholinguistic features to specifically investigate how reviewer communication differs for agent- versus human-authored PRs, by linking language use to collaboration dynamics and review outcomes.

## 6 Threats to Validity

**External Validity.** Our analyses rely on the AIDEV-pop dataset, a large-scale dataset capturing autonomous coding agent activity on GitHub repositories [17]. While AIDEV represents the largest publicly available dataset of agent-authored pull requests to date, it constitutes a single-source dataset. As such, our findings may reflect characteristics specific to the repositories, selection criteria, or agent activity captured within AIDEV. Although the dataset spans 2,807 repositories and multiple coding agents, results may not fully generalize to private repositories, smaller projects, alternative platforms, or future generations of autonomous agents.

**Construct Validity.** Our measures of PR characteristics (e.g., churn, cyclomatic complexity) serve as structural proxies rather than direct indicators of intrinsic code quality. Similarly, merge rate and time-to-merge capture observable integration outcomes but may not fully reflect downstream quality effects.

**Internal Validity.** As an observational study, we cannot infer causal relationships between authorship type and merge outcomes. Unobserved factors such as repository governance practices, contributor reputation, or prior interaction history may influence review decisions.

## 7 Conclusion and Future Work

This study provides large-scale empirical evidence that autonomous coding agents reshape not only development outcomes but also collaboration dynamics in open-source workflows. By comparing more than 40,000 pull requests across 2,807 repositories, we show that agent-authored contributions exhibit a consistent trade-off: they are integrated faster but are less likely to be merged overall. Crucially, this pattern is moderated by task type, with agents performing competitively in documentation-related tasks but underperforming in behavior-changing contributions.

Beyond integration outcomes, we observe clear socio-technical differences in how contributions are reviewed and discussed. Agent-authored pull requests attract more bot-generated comments and elicit more analytic, less socially oriented communication, whereas human-authored contributions generate more elaborative and socially engaged interactions. Psycholinguistic signals provide incremental explanatory power for merge outcomes, indicating that communication style meaningfully relates to integration decisions.

Overall, repository-level AI autonomy shapes both technical outcomes and collaborative dynamics. Autonomous agents are not merely code generators; they participate in social review processes that involve trust calibration and evaluative adaptation. Their impact is therefore fundamentally socio-technical.

Future research should examine how human–agent interaction patterns evolve as developers gain familiarity with autonomous contributors. Controlled studies could help isolate the causal mechanisms underlying acceptance differences, including the role of authorship perception and explanation quality, while also assessing the cognitive demands and mental load associated with human–AI collaboration. Finally, designing tools that improve transparency and explicitly communicate the rationale behind agent-generated changes may help reviewers better understand and evaluate these contributions.

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

Received 20 February 2007; revised 12 March 2009; accepted 5 June 2009

