# OpenReview forum: "When Code Authors Are Agents: A Large-Scale Study of Human–Agent Collaboration in Pull Requests"
_ACM.org/AIWare/2026/Conference — AIware 2026_

### Official Review · Reviewer_ajba · 2026-03-07

**Rating:** 3
**Confidence:** 4

**Review:**

This paper is generally well organized, and the main results are clearly communicated through tables and figures. The statistical tests used are standard for this type of empirical analysis. I found there are four main strengths:
1. **Timely topic**:  The paper addresses an emerging phenomenon: autonomous coding agents acting as first-class contributors in collaborative software development. The paper appropriately frames the problem as socio-technical rather than purely technical, recognizing that agents reshape not only code artifacts but also review interactions and collaboration dynamics. This framing aligns well with the goals of this conference.
2. **Substantial dataset with multi-agent coverage**:  The study analyzes over 40,000 pull requests across 2,807 repositories, spanning five distinct coding agents (Codex, Devin, Copilot, Cursor, and Claude Code). Compared with prior studies that often focus on a single tool or smaller samples, this scale and breadth strengthen the empirical basis of the findings and allow for more generalizable observations.
3. **Multi-dimensional analytical framework**: The organization around three complementary dimensions (integration outcomes, structural characteristics, and collaboration signals) provides a coherent and comprehensive analytical structure. This framework allows the study to move beyond simple merge-rate comparisons and examine how agent authorship affects both the technical properties of contributions and the interactional dynamics surrounding them.
4. **Psycholinguistic analysis**: The application of LIWC-based psycholinguistic analysis to study reviewer communication patterns is a distinctive and relatively uncommon addition in empirical software engineering research on AI-generated code. The finding that linguistic features provide statistically significant incremental predictive power for merge outcomes supports the authors' argument that collaboration dynamics extend beyond code artifacts.

However, I also want to point out the weakness that I want the authors can consider:

1. **Dataset construction introduces a systematic sampling asymmetry**:  The human and agent PR datasets are constructed using different repository selection thresholds: repositories with more than 100 stars for agent PRs versus more than 500 stars for human PRs. This means agent-authored PRs include contributions from smaller, potentially less mature repositories that are entirely excluded from the human sample. Since repository characteristics strongly influence review dynamics, merge behavior, and collaboration norms, this asymmetry introduces a systematic confound that could affect all three research questions. The authors should consider restricting comparisons to repositories where both human and agent PRs appear, or including repository-level controls such as project size, activity level, or fixed effects in the regression models.

2. **Limited control for confounding factors in regression models**: The regression analyses include several structural and linguistic predictors but do not control for a number of factors that are known to influence pull request outcomes. For example, repository-level characteristics, contributor experience, and prior interaction history between reviewers and contributors are not included in the models. These factors could plausibly affect both merge likelihood and review dynamics. As a result, some of the observed differences between human- and agent-authored PRs may reflect differences in project context rather than authorship alone. Incorporating additional contextual controls or hierarchical models that account for repository-level variation would strengthen the causal interpretation of the findings.

3. **Aggregation across heterogeneous agents obscures important variation**: The study aggregates PRs from five different coding agents (OpenAI Codex, Devin, GitHub Copilot, Cursor, and Claude Code) into a single "agent-authored" category. However, the paper itself acknowledges in the discussion that "variation across tools indicates that agent behavior is not uniform" and cites prior work showing that agentic PRs differ substantially across tools. Without per-agent descriptive statistics or the inclusion of agent identity as a factor in regression models, it is impossible to determine whether the observed patterns are consistent across agents or driven disproportionately by one or two tools. At minimum, reporting the distribution of PRs per agent and key outcome metrics stratified by agent would substantially improve interpretability.

4. **Insufficient transparency on task-type labeling**: The analysis depends heavily on task-type categories inherited from the AIDEV dataset, and while the paper lists the original labels and explains how they are grouped into four broader categories, it does not explain in this paper how the original labels were produced or validated. More detail on label provenance and reliability would be appreciated.

**Summary:**

This paper presents a large-scale empirical study of human–agent collaboration in GitHub pull requests. The authors analyze 40,214 pull requests across 2,807 repositories. The dataset includes 33,596 agent-authored PRs from five coding agents (Codex, Devin, Copilot, Cursor, Claude Code) and 6,618 human-authored PRs. The study compares agent- and human-authored PRs across three dimensions.
1. **Integration outcomes**: merge rate and time-to-merge
2. **Structural characteristics**: task type, change breadth, diff size, and cyclomatic complexity
3. **Collaboration signals**: commenter types (human vs bot) and psycholinguistic properties of review comments derived from LIWC

The authors report several findings. Human-authored PRs have a higher merge rate, while agent-authored PRs are merged significantly faster once accepted. Task type moderates these effects. Agents perform particularly well on documentation-related changes but underperform on behavior-changing contributions. The analysis also shows that agent-authored PRs attract a higher proportion of bot comments. Reviewer communication for agent PRs tends to be more analytic and less socially oriented. The authors also show that adding psycholinguistic features improves merge outcome prediction models.

---

> ### Author Response · Authors · 2026-03-19
>
> **1. Dataset construction introduces a systematic sampling asymmetry:** The original AIDEV release included a human-authored PR dataset; however, it lacked detailed information such as commit and review data, which are necessary for the analyses within the scope of our work. A more complete version of the human-authored PR dataset was provided to us directly by the dataset authors upon request and was extracted using a >500 stars threshold, whereas the original AIDEV dataset for agent-authored PRs uses a >100 stars threshold.
>
> We were aware of this discrepancy during the study design and proactively conducted analyses using matched thresholds (i.e., restricting agent-authored PRs to >500 stars). These analyses yielded results consistent with those reported in the paper, including (i) lower merge rates for agent-authored PRs, (ii) statistically significant differences (effect size 0.22), and (iii) faster time-to-merge for accepted PRs. As both the direction and significance of effects remained stable, we retained the full dataset to maximize statistical power and coverage. We will clarify this design decision and include the matched analysis as a robustness check
>
> **2. Limited control for confounding factors in regression models** We appreciate this important point. We agree that factors such as repository characteristics, contributor experience, and prior interaction history may influence pull request outcomes and review dynamics. In the current study, our models focus on structural and linguistic features, and we do not explicitly control for all contextual factors. As such, we acknowledge that some observed differences between human- and agent-authored PRs may reflect underlying project context rather than authorship alone.
>
> We will clarify this limitation in the paper and expand the discussion to explicitly acknowledge these potential confounding factors. We will also highlight the incorporation of richer contextual controls and hierarchical modeling (e.g., accounting for repository-level variation) as important directions for future work.
>
> **3. Aggregation across heterogeneous agents obscures important variation** We agree that aggregating across heterogeneous agents may obscure important variation. To improve transparency, we computed descriptive statistics stratified by individual agents, including the number of pull requests and merge rates per agent.
> The results show variability across agents, confirming that agent behavior is not uniform. However, the overall trend of lower merge rates relative to human-authored pull requests remains consistent across agents. As the primary goal of this work is to contrast human vs. autonomous authorship, we chose this aggregation for the main analysis.
>
> In the revision, we will include a summary table with per-agent statistics to improve interpretability while keeping the main focus on human-agent comparison.
>
> **4. Insufficient transparency on task-type labeling:** We appreciate the request for clarification. Task-type labels are inherited from the AIDEV dataset. In that dataset, pull requests are automatically classified into predefined categories based on the Conventional Commits specification, using LLM-based classification applied to pull request titles and commit messages.
>
> While this labeling process is described in the original dataset documentation, we agree that its provenance and reliability should be more clearly stated in our paper. We will add a brief description of the labeling process and include appropriate references to the original source to improve transparency.

---

### Official Review · Reviewer_7U3s · 2026-03-11

**Rating:** 2
**Confidence:** 5

**Review:**

Strengths:
------------------------------
+ The topic is highly timely and relevant. Studying autonomous coding agents at the PR level is clearly important for the SE community.
+ The paper asks a meaningful question that goes beyond code quality alone and looks at human-agent collaboration signals in real review workflows.
+ Novel psycholinguistic analysis
+ Clear implications for both research and practice
+ The paper is well-written, generally readable, and the main findings are straightforward to understand. Tables 2–5 communicate the key results clearly.

Weaknesses:
------------------------------------
- The paper presents itself as providing “the first large-scale empirical comparison of human and autonomous authorship in open-source pull requests,” but that claim is difficult to sustain against the recent literature it itself cites. Most notably, The Rise of AI Teammates in Software Engineering (SE 3.0) [17] already introduced the AIDev dataset and described AIDev-pop as a filtered subset of PRs from popular repositories, with comparisons against human PRs across task categories, acceptance rates, timing/latency, and structural characteristics. It reports 456,535 agentic PRs overall, an AIDev-pop subset over popular repositories, lower acceptance rates for autonomous agents than humans, documentation as a relative strength, faster review/merge patterns for some agents, and lower rates of structural complexity change in agentic code. Moreover, Watanabe et al. [37] already establish: (1) agent PRs have lower acceptance rates than human PRs overall, (2) agents perform better on documentation/testing tasks, (3) rejection is driven by coordination and scope misalignment rather than code defects, and (4) 54.9% of merged agent PRs require no modification. The paper must more explicitly frame its novelty against them, not just against general prior work.
-  The agent-to-human PR ratio is 33596:6618 (approximately 5:1). This severe imbalance is never justified, controlled for, or discussed as a threat.
- The human PR baseline is sampled from the same repositories as agent PRs but with a different star threshold (>500 vs >100 stars) and without matching on time period, developer experience, task type distribution, or PR size. This means the comparison is not between humans and agents doing the same kinds of work in the same contexts, but between an opportunistic mixture of whatever agent PRs exist and a convenience sample of human PRs. Differences in merge rates, complexity, or linguistic patterns may reflect repository governance culture, contributor seniority, or task distribution differences rather than authorship type per se. The paper's causal language ("agent-authored PRs exhibit lower merge rates") overstates what an unmatched observational comparison can support. The paper acknowledges observational limits in threats to validity, but that is not enough here because many headline interpretations rely on differences that may reflect who submits what kind of PR under what conditions, not authorship alone.
- The LIWC component is interesting, but the actual effects in Table 4 are mostly small. Many differences are statistically significant because the dataset is large, yet the reported standardized effects are small or negligible.
- Five distinct agents are included (OpenAI Codex, Devin, GitHub Copilot, Cursor, Claude Code), yet virtually all analyses collapse them into a single "agent" category. Li et al.'s AIDev paper itself notes that "variation across tools indicates that agent behavior is not uniform", and Ogenrwot and Businge [22] show agent behavior varies significantly across tools. Treating all five agents as a homogeneous "agent" category is a significant analytical weakness. Merge rates, structural complexity, and communication patterns almost certainly differ substantially across Devin (a full autonomous SE agent) vs. GitHub Copilot (closer to an assistant tool) vs. Claude Code. Without per-agent breakdowns, the aggregate findings risk being dominated by the most-represented agent(s) in the dataset, which appears to be Codex/Copilot given the dataset composition.
- The paper repeatedly argues that the impact of coding agents is “fundamentally socio-technical". That may be true, but the paper does not fully develop what that means analytically other than merge outcomes differ, reviewer language differs, and bot/human commenting mixes differ.
- No replication package

**Summary:**

This paper presents a large-scale empirical study comparing 40,214 pull requests (33,596 agent-authored and 6,618 human-authored) across 2,807 GitHub repositories, examining three dimensions: integration outcomes, structural/patch characteristics, and collaboration and communication signals using psycholinguistic (LIWC) analysis. The study uses the AIDEV-pop dataset and covers five autonomous coding agents: OpenAI Codex, Devin, GitHub Copilot, Cursor, and Claude Code. The central finding is a socio-technical trade-off: agent-authored PRs are merged faster but at lower overall rates, with task type moderating this effect, and reviewer communication being measurably more analytic and less socially engaged for agent-authored contributions.

---

> ### Author Response · Authors · 2026-03-19
>
> **1. Novelty claim overlaps with prior AIDEV-based studies** We thank the reviewer for this important observation and agree that our novelty claim should be more carefully positioned with respect to prior work on the AIDEV dataset and related studies.
> We will revise the claim to more accurately reflect existing literature, including SE 3.0 and Watanabe et al., which have already established key differences between human- and agent-authored pull requests across acceptance rates, task types, and structural characteristics. In the final version, we will explicitly frame our contribution as complementary to these works. Specifically, our novelty lies in integrating psycholinguistic analysis of review communication with structural and outcome-based comparisons, providing insight into how agent-authored contributions are discussed and evaluated by human reviewers, rather than in being the first to perform large-scale comparisons.
>
>
> **2. The agent-to-human PR ratio is 33596:6618 (approximately 5:1).** We acknowledge the imbalance between agent- and human-authored pull requests (33596:6618, ~ 5:1), which reflects the underlying dataset rather than a design choice. During our initial analysis, we also examined a matched subset using the same threshold (>500 stars), resulting in a less imbalanced ratio (~2:1). The observed trends remained consistent with those reported in the paper. Based on this, we decided to retain the full dataset to maximize statistical power and coverage. We will discuss class imbalance as a threat to validity in the revised manuscript.
>
> **3. 	Unmatched human vs. agent PR comparison (> 500 vs. > 100)** The original AIDEV release included a human-authored PR dataset; however, it lacked detailed information such as commit and review data, which are necessary for the analyses within the scope of our work. A more complete version of the human-authored PR dataset was provided to us directly by the dataset authors upon request and was extracted using a >500 stars threshold, whereas the original AIDEV dataset for agent-authored PRs uses a >100 stars threshold.
> We were aware of this discrepancy during the study design and proactively conducted analyses using matched thresholds (i.e., restricting agent-authored PRs to >500 stars). These analyses yielded results consistent with those reported in the paper, including (i) lower merge rates for agent-authored PRs, (ii) statistically significant differences (effect size 0.22), and (iii) faster time-to-merge for accepted PRs. As both the direction and significance of effects remained stable, we retained the full dataset to maximize statistical power and coverage. We will clarify this design decision and include the matched analysis as a robustness check.
>
> **4. 	The LIWC component is interesting, but the actual effects in Table 4 are mostly small.** We will revise the manuscript to explicitly distinguish statistical significance from practical significance and avoid over-interpreting small effects.
>
> **5. 	Aggregating heterogeneous agents into a single category may obscure important per-agent differences.** During the initial analysis, we computed per-agent statistics (e.g., PR counts, merge rates), which showed variability while preserving the overall trend of lower merge rates compared to human-authored PRs. However, drawing meaningful per-agent conclusions would require a more in-depth technical comparison of agent capabilities and usage contexts, which is beyond the scope of this work. For this reason, we did not include per-agent results. In the revision, we will add a summary table to improve transparency while maintaining the focus on human–agent comparison.
>
> **6. 	Socio-technical framing is not sufficiently developed or operationalized analytically** We appreciate this observation. By “socio-technical,” we refer to the interplay between technical characteristics of contributions (e.g., structure, scope, merge outcomes) and social aspects of the review process (e.g., reviewer communication and interaction patterns).
> In the revision, we will add a brief clarification to more explicitly connect our existing measures to this perspective, namely: structural metrics as the technical dimension, reviewer communication as the social dimension, and merge outcomes/timing as their interaction. Our intent is not to introduce a new analytical framework, but to make this interpretation of the existing results more explicit.
>
> **7. No replication package** We recognize the importance of providing a replication package and appreciate the reviewer’s comment. The submission system did not offer an option to include it, and since the AIDEV dataset is already publicly available, we had initially planned to release ours after submission. In response, we have now prepared our own replication package and made it available through this anonymized link: https://anonymous.4open.science/r/Human-Agent-Collaboration-C973/README.md

---

### Official Review · Reviewer_HEMK · 2026-03-12

**Rating:** 3
**Confidence:** 4

**Review:**

# Importance

The work addresses a timely topic. Understanding how agent contributions are reviewed and integrated by human teams is relevant to SE. The framing of agents as socio-technical participants rather than mere code generators is valuable.

# Originality
The main novelty lies in applying LIWC to study reviewer communication shifts. While merge rate and time-to-merge comparisons have appeared in concurrent work (Watanabe et al. [33], Pham and Ghaleb [26]), the linguistic dimension is a meaningful addition. However, the claim of "the first large-scale empirical comparison" should be stated more precisely, given that multiple MSR 2026 Mining Challenge papers [7, 12, 19, 22, 24, 26, 36] perform similar comparisons on the same AIDEV dataset.

# Soundness

**Sampling bias.** Agent PRs are filtered at >100 stars while human PRs at >500 stars. This means human PRs exclusively come from more mature repositories with potentially stricter review standards, confounding all three RQs. The authors should replicate key analyses on matched repository sets.

**Bot comments in LIWC analysis.** Table 5 shows 61.46% of comments on agent PRs are bot-generated. If bot comments were not excluded before the psycholinguistic analysis, the LIWC differences likely reflect bot vs. human language characteristics rather than genuine shifts in reviewer behavior. This could undermine the core contribution of RQ3 and must be clarified.

**Small effect sizes (RQ1).** The merge rate difference (77.2% vs. 71.5%, Cramér's V = 0.05) is negligible by conventional standards. With N > 40,000, trivially small differences reach significance. The paper should more carefully distinguish statistical from practical significance.

**Time-to-merge needs deeper investigation.** Agent PRs merging in ~19h raises questions: Is this driven by auto-merge bots? Concentrated in specific repositories? Are PR submitters also maintainers with merge privileges? Combined with the RQ3 finding that agent PRs attract more bot comments, this suggests a possible automated review-and-merge pipeline — effectively agents reviewing agents. Stratifying by repository characteristics (e.g., star count, governance model) would strengthen the analysis.

**Agent heterogeneity.** The five agents (Codex, Devin, Copilot, Cursor, Claude Code) differ substantially. The paper cites prior work noting "agent behavior is not uniform" but provides no per-agent breakdown. At minimum, descriptive statistics per agent should be included.

**Metric clarity.** "Breadth of change" is operationalized as file/hunk counts but could be misread as cross-module scope — consider renaming (e.g., "dispersion of change"). "Diff size" is said to reflect "logical changes rather than raw line counts" without specifying how these are counted. The grouping of task labels into four categories (e.g., `feat` + `fix` + `perf`) also lacks explicit justification; `perf` may be more similar to `refactor` than to `feat`.

**LIWC predictive power.** The significant likelihood ratio test (χ²(2) = 49.28) is expected at N > 40,000 for any weakly informative variable. Without reporting AUC change or pseudo-R², the claim that "communication style carries explanatory power" is difficult to evaluate.

# Presentation

Figures 1 and 2 are blurry. Figure 1 contains what appears to be a Gemini watermark in the lower-right corner. Please redraw and upload vector graphics.
The related work could benefit from broader coverage of papers in SE venues on human-AI collaboration, e.g., see:

Tao Xiao, Hideaki Hata, Christoph Treude, Kenichi Matsumoto. "Generative AI for Pull Request Descriptions: Adoption, Impact, and Developer Interventions." Proc. ACM Softw. Eng. 1, FSE, Article 47 (2024), 1043–1065.

Junda He, Christoph Treude, David Lo. "LLM-Based Multi-Agent Systems for Software Engineering: Literature Review, Vision, and the Road Ahead." ACM Trans. Softw. Eng. Methodol. 34(5): 124:1-124:30 (2025).

# Questions:

1. Why do the star thresholds differ for agent (>100) and human (>500) PRs?
2. Were bot comments excluded from the LIWC analysis? If not, how do you ensure observed differences reflect human reviewer behavior?
3. Have you investigated whether fast agent merge times are driven by automated pipelines or concentrated in specific repositories?

**Summary:**

# Paper Summary
This paper conducts a large-scale empirical study comparing 40,214 pull requests (33,596 agent-authored, 6,618 human-authored) across 2,807 GitHub repositories. It examines three dimensions: integration outcomes, structural characteristics, and collaboration signals (including LIWC-based psycholinguistic analysis). The key finding is a socio-technical trade-off: agent PRs merge faster but have lower merge rates, moderated by task type. Psycholinguistic features provide incremental predictive power for merge outcomes.

# Strength
- A novel socio-technical perspective that goes beyond purely technical evaluation of agent-generated code.
- Task-type moderation analysis reveals meaningful interaction effects rather than treating all PRs as homogeneous.
- The introduction of LIWC-based psycholinguistic analysis to study reviewer behavior toward agent PRs is creative and relatively unexplored.

# Weakness
- Unequal sampling criteria for agent vs. human PRs introduce systematic bias.
- Unclear whether bot comments were excluded from the psycholinguistic analysis.
- Five agents are analyzed in aggregate without a per-agent breakdown.
- Missing qualitative analysis to explain why observed patterns exist.

---

> ### Author Response · Authors · 2026-03-19
>
> **Q1. Star thresholds differences for agent (>100) and human (>500) PRs?** The original AIDEV release included a human-authored PR dataset; however, it lacked detailed information such as commit and review data, which are necessary for the analyses within the scope of our work. A more complete version of the human-authored PR dataset was provided to us directly by the dataset authors upon request and was extracted using a >500 stars threshold, whereas the original AIDEV dataset for agent-authored PRs uses a >100 stars threshold.
> We were aware of this discrepancy during the study design and proactively conducted analyses using matched thresholds (i.e., restricting agent-authored PRs to >500 stars). These analyses yielded results consistent with those reported in the paper, including (i) lower merge rates for agent-authored PRs, (ii) statistically significant differences (effect size 0.22), and (iii) faster time-to-merge for accepted PRs. As both the direction and significance of effects remained stable, we retained the full dataset to maximize statistical power and coverage. We will clarify this design decision and include the matched analysis as a robustness check.
>
>
> **Q2. Bot comments excluded from the LIWC analysis?** Yes. Bot-generated comments were excluded prior to the LIWC analysis. All psycholinguistic analyses were conducted exclusively on human-authored comments, ensuring that the results reflect human reviewer behavior. The analysis of bot comments in the paper serves a separate purpose, namely, to characterize the distribution of commenter types. We will clarify this distinction in the revision.
>
> **Q3. Are fast agent merge times driven by automation or specific repositories?** We have not conducted a dedicated analysis to attribute faster merge times to automated pipelines or repository-specific effects. However, the effect appears consistent across the dataset, suggesting it is not driven solely by a small subset of repositories. At the same time, the higher prevalence of bot activity indicates that automation may play a role. We will expand the discussion to explicitly consider automation and repository-level characteristics (e.g., size, activity level, governance) as plausible explanations and outline this as future work.
>
> **Additional clarifications.**
> **Novelty:** We agree and will revise our novelty claim to better situate the work within concurrent MSR 2026 studies. Our contribution lies in integrating psycholinguistic analysis with structural and outcome-based comparisons of pull requests.
>
> **Small effect sizes (RQ1):** We appreciate this point. We will revise the paper to more clearly distinguish between statistical and practical significance by adding to the discussion that, while the differences in merge rates are statistically significant, the small effect size indicates that authorship alone explains only a modest portion of the variance in integration outcomes.
>
> **Agent heterogeneity:** We also acknowledge agent heterogeneity. We computed per-agent descriptive statistics (e.g., PR counts, merge rates), which show variability across agents while preserving the overall trend. However, without accounting for factors such as differences in agent capabilities and usage contexts, it is difficult to draw meaningful conclusions. We will include a summary table in the revision to improve transparency.
>
> **Metric clarity:** Regarding metrics, we will improve clarity by renaming “breadth of change” to “dispersion of change” and explicitly defining “diff size” as a measure of logical change magnitude. Specifically, we aggregate additions and deletions per file across commits, take the maximum per file, and sum across files to reduce double counting and better reflect effective changes. We will also better justify the grouping of task labels.
>
> **LIWC contribution:** We now report model fit metrics for LIWC (pseudo R² = 0.036), indicating modest explanatory power. We will clarify that communication style provides incremental, but not dominant, predictive value.
>
> **Presentation:** We will replace figures with higher-quality versions and incorporate the suggested references into the related work.